# Mapping the evidence on outcomes of childhood out-of-home care: A scoping review of reviews

Richmond Opoku[1,2*], Natasha Judd[3], Katie Cresswell[3], Michael Parker[1], Michaela James[1,2], Jonathan Scourfield[4], Karen Hughes[3,5], Jane Noyes[1,6], Dan Bristow[7], Evangelos Kontopantelis[8], Sinead Brophy[1,2,9], Natasha Kennedy[1,9]

1 The Centre for Population Health (CPH), Medical School, Swansea University, Wales, United Kingdom, 2 Administrative Data Research Wales (ADR-Wales), Swansea, United Kingdom, 3 Public Health Collaborating Unit, School of Health Sciences, College of Medicine and Health, Bangor University, Wrexham, Wales, United Kingdom, 4 Children's Social Care Research and Development Centre (CASCADE), School of Social Sciences, Cardiff University, Cardiff,, United Kingdom, 5 Policy and International Health, World Health Organization Collaborating Centre on Investment for Health and Wellbeing, Public Health Wales, Wrexham, Wales, United Kingdom, 6 School of Health Sciences, Bangor University, Wales, United Kingdom, 7 Wales Centre for Public Policy, Cardiff University, Wales, United Kingdom, 8 Division of Informatics, Imaging and Data Sciences, The University of Manchester, Manchester, United Kingdom, 9 Health Data Research United Kingdom, Wales (HDRUK Wales), Swansea, United Kingdom

* 2349105@swansea.ac.uk

## Abstract

### Background

Children placed in out-of-home care in high-income countries face complex challenges due to exposure to adverse childhood experiences and systemic disadvantages. While research on their outcomes has grown, the evidence base remains fragmented. An overview of review-level evidence was conducted to identify patterns, gaps, and priorities for future research and practice.

### Methods

A scoping review of reviews was conducted. Peer-reviewed review articles published between January 2013 and July 2024 were identified through searches in databases including EBSCOhost, ProQuest, Cochrane Database of Systematic Reviews, and Epistemonikos. Eligible reviews focused on childhood out-of-home care experience and reported outcomes for care-experienced individuals (assessed either in childhood or adulthood) and/or associated factors. Outcomes were categorised under the following domains: Health and Emotional Wellbeing (HEW), Physical and Legal Security (PLS), Education and Learning (EL), Living Standards and Social Wellness (LSSW), and Identity and Civic Participation (ICP). Factors were classified across multiple levels, including individual child-level, socio-relational-level, community-level, system-level, and other factors.

**Data availability statement:** All relevant data are within the manuscript and its Supporting Information files.

**Funding:** This study is funded by the NIHR Health and Social Care Delivery Research Programme (NIHR156826 - CARELINK Wales - Comprehensive Analysis of Risk factors and outcomes for vulnerable children through LINKed Welsh Data), UK. The views expressed are those of the author(s) and not necessarily those of the NIHR or the Department of Health and Social Care. Additional support was provided by the Economic and Social Research Council–Administrative Data Research UK (ESRC-ADR UK) through a PhD studentship. The funders had no role in study design, data collection and analysis, decision to publish, or preparation of the manuscript.

**Competing interests:** The authors have declared that no competing interests exist.

## Results

A total of 77 reviews were included, spanning diverse methodologies and contexts. Research was concentrated in domains such as HEW and LSSW, with indicators such as mental and emotional health and attachment and behaviour functioning receiving substantial attention. Conversely, key gaps were identified, including the limited reporting of ICP outcomes (e.g., identity and self-respect). System-level factors, such as care quality and placement type, were most frequently reported across outcome domains and indicators. Individual child level and socio-relational-level factors were consistently highlighted, while community-level factors were largely underrepresented.

## Conclusion

Future research should target gaps in underexplored outcome domains like ICP and indicators such as bullying, mortality, and educational readiness. Community-level factors warrant more attention as they play a significant role in supporting transitions to independence and social integration.

## Background

In most high-income countries, current laws and policies emphasise placing children who cannot remain with their parents into out-of-home care (OHC), with a preference for placements with extended family members or close family friends (kinship care) [1–3]. We define out-of-home care (OHC) as any short- or long-term alternative living arrangement for children who can no longer live with their parents due to safety or welfare concerns. Some argue that kinship care arrangements are associated with better developmental outcomes compared to placements in unrelated foster care or institutional settings [4,5]. However, children in any form of OHC remain a particularly vulnerable group, with complex needs and facing numerous challenges. These challenges often stem from adverse childhood experiences (e.g., abuse and neglect) and systemic disadvantages encountered prior to, during, and sometimes after their time in care [4,6–10].

Over the past decades, increasing attention has been given to the experiences and outcomes of children in OHC, reflected in a growing body of research, including several literature reviews [e.g., 11–16]. These reviews synthesise evidence across diverse contexts, but the research landscape remains fragmented, with studies varying in focus and methodological approach. With this in mind, a comprehensive map of studied outcomes and their associated factors would provide a clearer understanding of the breadth of available evidence, help identify gaps, and highlight priority areas for future research and intervention. Accordingly, the research question guiding this study was: what outcomes of out-of-home care (OHC) have been examined in existing reviews, and what factors have been studied in relation to these outcomes?.

## Materials and methods

### Review design

A scoping review of reviews was determined to be the most suitable approach for capturing the breadth of this extensive research area, systematically mapping the existing evidence, and identifying gaps in the literature. The protocol for the review is available at https://doi.org/10.17605/OSF.IO/G7D5J. We followed the six stages recommended by Arksey and O'Malley [17] and used the principles of Framework synthesis [18] for extracting relevant data. Reporting followed the Preferred Reporting Items for Systematic Reviews and Meta-Analyses (PRISMA) extension for scoping reviews [19].

### Search strategy

A search of academic databases was conducted to identify peer-reviewed review articles. The databases included EBSCOhost (searching MEDLINE, APA PsycArticles, APA PsycInfo, Education Research Complete, and CINAHL Ultimate), ProQuest (covering ASSIA, Criminal Justice Database, Education Database, and Social Science Database), Cochrane Database of Systematic Reviews, and Epistemonikos. The initial search strategy was developed by R.O. and underwent iterative refinement following inputs from J.N., K.H., S.B., and Patient and Public Involvement (PPI) participants. The full search strategy and the procedures to select studies are described in a related review of reviews [20].

### Eligibility criteria

A review was included if it met the following criteria:

- Reported any outcome(s) for individuals placed in OHC during childhood (i.e., under 18 years of age), assessed either in childhood or adulthood, and/or associated factors.

- Included quantitative, qualitative, or mixed-methods studies with a documented search strategy.

- Published from January 2013 to July 2024, available in full-text and written in English.

- Focused on global or high-income country contexts. The decision to focus on high-income countries was based on the following considerations:

  - These countries are more likely to have established and formalised systems of fostering, kinship, and residential care, offering a clearer basis for cross-study comparisons.

  - The review aimed to inform policy and research priorities within high-income contexts.

A review was excluded if it:

- Focused exclusively on children with learning disabilities, those in inpatient psychiatric care, young offender institutions, adoption, or specialist centres for mothers and children. These contexts often involve distinct care pathways, interventions, and outcome profiles that differ meaningfully from the broader looked-after population, and were thus beyond the scope of this review.

- Was non-empirical or exclusively focused on literature from low- and middle-income countries or postgraduate theses, books, or grey literature.

### Charting the data

Data extraction was carried out independently by three reviewers (R.O., K.C., and N.J.), while two supervising reviewers (M.J. and S.B.) verified the extracted data to minimise errors. The following information was collected from each included review: author, year of publication, title, type of review, number and type of included studies, countries covered by the

included studies, review time frame, population, analytical approach, aims/objectives, findings on factors associated with care entry, review authors' interpretations, and quality assessments of the included studies. It is common practice in studies mapping evidence on broad topics to limit data extraction to the abstract section [21–25]. However, data extraction for this review was extended to include the full text to minimise the risk of underestimating the number of reviews reporting relevant outcomes and factors [26].

## Collating, summarising, and reporting results

The large volume of records, combined with the broad scope of the topic and diverse outcome measures, makes synthesising and reporting specific findings impractical [25,26]. However, a descriptive analysis of study characteristics and outcome categories enabled a comprehensive mapping the literature. One reviewer (R.O.) categorised the outcomes of children in care using the domains and indicators of the Equality Measurement Framework (EMF), developed by the UK Equality and Human Rights Commission [27]. Adaptations were made to account for the EMF not being specifically designed for children in OHC. Some domains were merged, while additional indicators emerged from the data, resulting in the five outcome domains described in Table 1. Details on the development of outcome categories are available in S1 Table. Additionally, factors were categorised using the modified ecological model [7] as: *individual child level* (child characteristics), *socio-relational level* (family, peers, and carer relationships), *community level* (schools, neighbourhoods, and local environments), *system level* (services, policies, and organisational practices), and *other factors* (influences not captured by the preceding levels, such as study characteristics in meta-analyses). The term "factor(s)" is used broadly to encompass what influences outcomes – both risk and protective factors. The inclusion of factors was not limited to variables with statistically significant associations with the outcomes but also included themes identified in reviews that synthesised qualitative data.

We treated outcomes and factors as non-mutually exclusive analytical categories. Thus, whenever a review covered multiple outcomes or factors, it was assigned to all of them rather than just one. Each review was examined in full, and outcomes/factors were assigned whenever a review contained explicit material relevant to that category (e.g., reported outcomes, statistical associations, or qualitative themes). A single review could therefore contribute to multiple outcomes or factors, resulting in counts that legitimately exceed the number of included reviews. Coding occurred in two stages. First, R.O. mapped all reported outcomes and factors to the adapted EMF domains and ecological levels using a rule-based decision protocol (i.e., assignment only when directly supported by review content). Second, N.J. independently audited the outcome/factor assignments for consistency, conceptual accuracy, and adherence to coding rules. This procedure ensured that mapping reflected the full scope of each review rather than forcing reviews into a single dominant category.

**Table 1. Categorisation of outcome domains and indicators.**

| Outcome domain | Outcome indicators |
| --- | --- |
| Health and Emotional Wellbeing (HEW) | Mental and emotional health; Healthy living behaviours; General health and wellbeing; Physical health and disability; Service access and utilisation; Reproductive and sexual health; Suicidality and self-harm; All-cause mortality |
| Physical and Legal Security (PLS) | Offending behaviours; Maltreatment, abuse, and neglect; Arrests, referrals and convictions; General safety and risk; Incarceration and imprisonment; Victim of violent crime; Experienced bullying |
| Education and Learning (EL) | Educational attainment; Cognitive and academic performance; School engagement; Educational readiness and access |
| Living Standards and Social Wellness (LSSW) | Attachment and behaviour problems; Care related experiences; Employment and labour; Income, deprivation and poverty; Housing and accommodation; Adjustment and transition to adulthood |
| Identity and Civic Participation (ICP) | Identity and self-respect; Participation and influence |

## Public involvement

The review of reviews was undertaken with contributions from the Welsh National Centre for Population Health and Wellbeing Research public and patient involvement (PPI) group at multiple stages, including grant development, review conduct, and interpretation of results. This engagement helped to ensure that the research questions, methodological approach, and interpretation of findings were grounded in and informed by the lived experiences of individuals affected by the child welfare system. Public involvement activities focused on two principal stakeholder groups: children and young people aged 15–25 with care experience, engaged through CASCADE Voices, a research advisory group for care-experienced young people, and parents with experience of having a child removed from their care.

## Results

### Review selection and characteristics

Fig 1 presents detailed data on the review selection process. Of the 711 references initially imported, 656 were screened by title and abstract, leading to 143 full-text assessments. Finally, 77 reviews met the eligibility criteria for inclusion. Among these reviews, 55 (71.4%) were systematic reviews [11,12,16, 28–72], of which 17 incorporated a meta-analysis [5,16,58,60–67,73–78]. Six (7.8%) were scoping reviews [13,14,79–82], and 16 (20.8%) were other review types [15,83–97]. The reviews included primary studies from at least 48 countries, with most conducted in the USA, followed by the UK, Canada, Australia, and Sweden. The primary studies spanned the years 1972–2022. Over 60% of the reviews assessed either the risk of bias or the quality of the included primary studies. The full characteristics of the included reviews are presented in Table 2.

### Patterns in OHC outcomes reported in the included reviews

Publication activity on OHC outcomes increased over time before declining in recent years (Fig 2). *Living Standards and Social Wellness* was the most frequently reported outcome domain, while *Identity and Civic Participation* and *Physical and Legal Security* were least reported. *Health and Emotional Wellbeing* showed a steady level of research activity across the period, and *Education and Learning* gained attention from around 2017, followed by more modest levels afterward.

The distribution of reviews across outcome indicators shows clear differences in research attention across domains (Fig 3). Within *Health and Emotional Wellbeing*, mental and emotional health received the greatest focus, while outcomes such as mortality were least examined. In *Physical and Legal Security*, research centred mainly on offending behaviours, with limited attention to victimisation, safety, and bullying. For *Education and Learning*, educational attainment dominated the evidence base, whereas readiness and access were less explored. In *Living Standards and Social Wellness*, attachment and behaviour functioning received the most attention, while housing outcomes were less frequently examined. Within *Identity and Civic Participation*, identity and self-respect were more commonly reported than participation and influence. Overall, these patterns point to strong concentration on mental health and attachment, alongside gaps in areas such as bullying, mortality, and early educational processes

### Factors associated with the outcomes of OHC

**Factors reported across outcome domains.** Across outcome domains, risk and protective factors were most often reported at the system, socio-relational, and individual child levels (Fig 4). For *Health and Emotional Wellbeing*, system-level influences were most prominent, followed by individual and socio-relational factors. *Physical and Legal Security* outcomes were mainly linked to socio-relational and system-level factors, with fewer references to individual child influences. For *Education and Learning*, socio-relational factors were most frequently reported, followed by system-level and individual factors. *Living Standards and Social Wellness* showed strong emphasis on both system-level and socio-relational influences, with individual-level factors also commonly reported. For *Identity and Civic Participation*, socio-relational and other contextual factors were most often cited, with fewer system-level and individual-level influences.

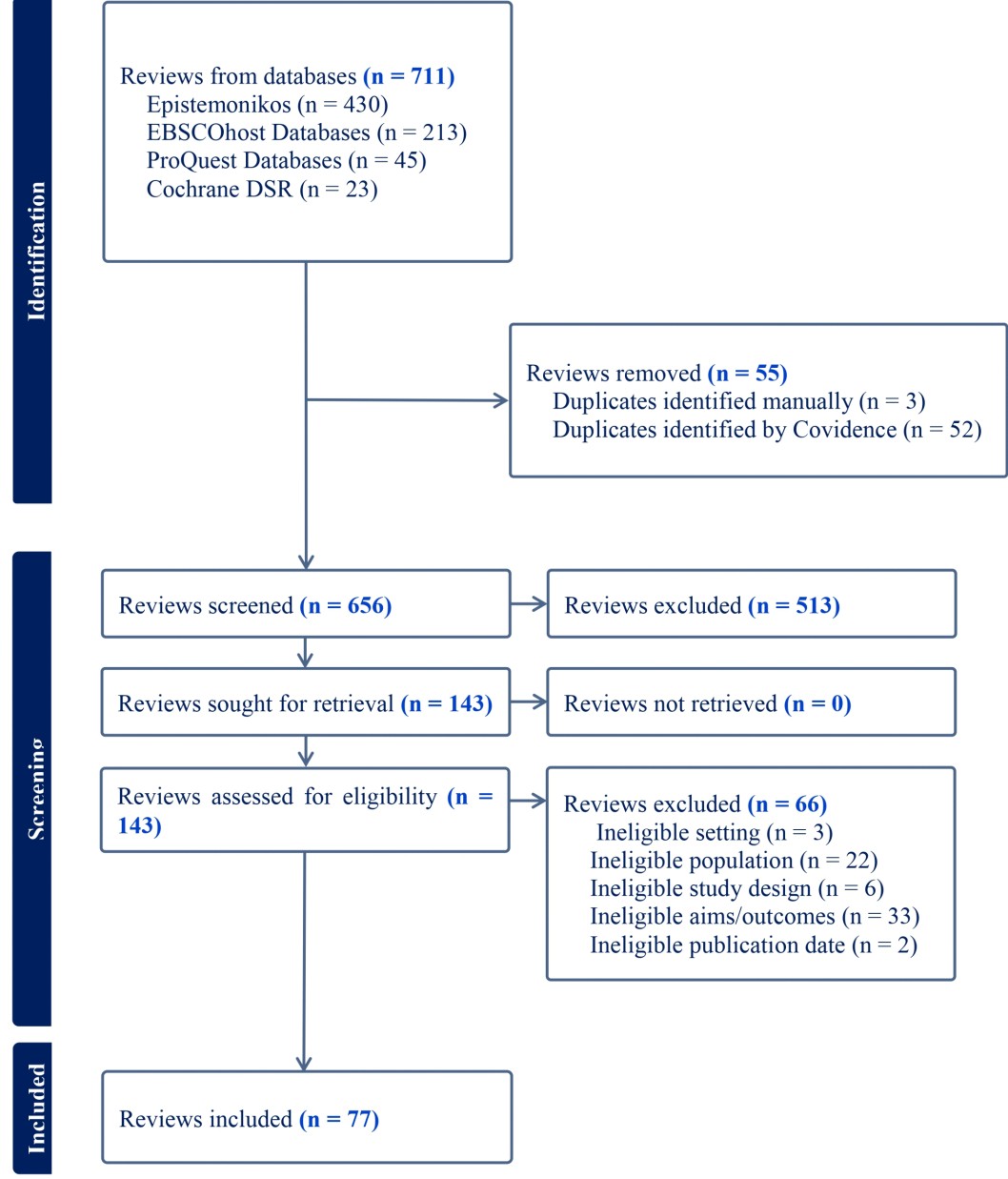

**Fig 1. PRISMA flow diagram illustrating the review selection process.**

## Health and emotional wellbeing

Within the *Health and Emotional Wellbeing* domain, most outcome indicators were primarily discussed in relation to system-level and individual child-level factors, with socio-relational influences also reported for several indicators (Fig 5). Mental and emotional health was mostly linked to system and socio-relational factors, while suicidality and self-harm were mainly discussed in relation to system-level influences. Mortality outcomes were largely examined through individual child-level factors, with little attention to social or community factors. Physical health and disability were also mainly linked to system-level factors. Reproductive and sexual health outcomes were most often related to socio-relational influences,

**Table 2. Characteristics of included reviews.**

| Study | Type of review | Number of studies included (*) | Type of included studies | Location of included studies | Review time frame | Population included |
|---|---|---|---|---|---|---|
| Welch [88] | Selective review | 90 (90) | Qa, Q1, Md | U.S.A = 54, U.K. = 20, Canada = 10, Australia = 4, China = 1 and The Netherlands = 1 | 1998–2013 | Children and young people in foster care and adoption |
| Geiger [82] | Scoping review | 51(51) | Q1, Qa & Md | US = 50, Europe = 1 | 1997-2015 | Foster care alumni in post-secondary education |
| Goding [48] | Systematic review | 11(11) | Qa & Md | UK | 2010-2019 | Children and young people in care |
| Batty [73] | Systematic review & meta-analysis | 14 (14) | Q1 | Finland = 4, Sweden = 4, UK = 3, Canada = 1, USA = 1, and Australia = 1 | 1995-2022 | Children (up to 18 years) exposed to temporary out-of-home care |
| Phillips [79] | Scoping review | 23(23) | Q1, Qa | USA = 11, UK = 3, Australia = 2, Canada = 2, Israel = 2, Spain = 1, and Sweden = 1 | 2006 - 2021 | Care-leavers, defined as any person who was under state care when they crossed the age of eligibility for being under state care. |
| Zabern [89] | Literature review | 17 (17) | Q1, Qa, Md | n.r | 2003 - 2016 | Children in out-of-home care. |
| Washington [59] | Systematic review | 40 (40) | Q1 | USA = 40 | 2010 - 2016 | Children residing in foster or kinship care. |
| Yoon [68] | Systematic review | 15 (15) | Q1 | n.r | 1985 - 2017 | Current or former youth in out-of-home care. |
| Tang [11] | Systematic review | 13 (6) | Q1 | USA = 4, Finland = 4, UK = 3, Australia = 1, and Luxembourg = 1. | 2009 - 2022 | Children <18 years of age exposed to early institutionalization, placement into foster care, parental incarceration, separation when parents migrate for economic reasons, separation due to asylum and war. |
| Kääriälä [69] | Systematic review | 20 (20) | Q1 | Sweden = 12, Finland = 5, Norway = 2, and Denmark = 1 | 2001 - 2016 | Populations born in 1965 or later in Denmark, Finland, Norway, or Sweden; reported any outcome of OoHC beginning at age 18 or later; compared a population that was placed in OoHC with a population that was never in OoHC during their childhood or adolescence. |
| Day [70] | Systematic review | 48 (48) | Qa | n.r | 1987-2016 | Resource parents who care for older youth |
| Häggman-Laitila [95] | Integrative review | 16 (16) | Q1, Qa | USA = 11, Australia = 2, Finland = 1, Israel = 1, UK = 1 | 2011 - 2018 | Care leavers who have left the foster care |
| O'Higgins [71] | Systematic review | 39(39) | Q1 | USA = 24, Canada = 5, UK = 5, Australia = 4 and Sweden = 1 | 1992 - 2014 | School-age children in foster or kinship care. |
| Cassarino-Perez [58] | Meta-analysis | 24 (24) | Q1 | USA = 14, UK = 1, Spain = 1, Sweden = 1 | 2006 - 2015 | Young people aging out of care (between 16 and 30 years old) |
| Quarmby [80] | Scoping review | 7(7) | Q1, Qa | USA = 3, UK = 2 and Norway 2 | 1998 - 2013 | Children and young people living in or leaving care |
| Häggman-Laitila [72] | Systematic review | 21 (21) | Qa | UK = 5, USA = 4, Canada = 2, Ghana = 2, India = 1, Ireland = 1, Norway = 1, Republic of Korea = 1, Romania = 1, South Africa = 1, Sweden = 1, Zimbabwe = 1 | 2010 - 2017 | Young people leaving foster care |
| Doab [28] | Systematic review | 11(11) | Q1, Qa | USA | 2007-2012 | Mothers with Substance Use Disorders (SUD) |
| Strijbosch [60] | Meta-analysis | 19 (19) | Q1 | North America = 15, Western Europe = 4 | 1987 - 2012 | Children and youth between 4 and 17 years old who are in care |

*(Continued)*

| Study | Type of review | Number of studies included (*) | Type of included studies | Location of included studies | Review time frame | Population included |
|---|---|---|---|---|---|---|
| Hurry [81] | Scoping review | 28(28) | Q1 | UK = 28 | 2001-2021 | Looked after children up to the age of 18 (extended to 22 years if the LAC was still in full time education) |
| Loomis [29] | Systematic review | 18(12) | Q1 | USA = 18 | 1999-2017 | Older youth currently or formerly in foster care (13 years+) |
| Townsend [30] | Systematic review | 11 (11) | Qa, Md | USA = 7, UK = 2, Canada = 1, Germany = 1 | 2003 - 2018 | Children currently in care and youth/adults who had previously been in care. |
| Winokur [31] | Systematic review | 102(102) | Q1, Qa | USA = 89 | 1991 - 2011. | Children and youth less than 18 years who were removed from the home for abuse, neglect, or other maltreatment and subsequently placed in kinship care. |
| Johnson [32] | Systematic review | 46(46) | Qa, Q1, Md | USA = 46 | 1990 - 2018 | Undergraduate Youth formerly in foster care (YFFC) |
| Bronsard [74] | Systematic review & meta-analysis | 8 (8) | Q1 | UK = 3, USA = 2, France = 1, Germany = 1, Norway = 1 | 1996 - 2013 | Children and adolescents involved in the child welfare system |
| Goemans [61] | Meta-analysis | 29 (29) | Q1 | USA = 15, The Netherlands = 4, Australia = 2, Croatia = 1, Iraq = 1, Norway = 1, Northern Ireland = 1, Scotland = 1, Canada = 1, UK = 1, Belgium = 1 | 1978 - 2013 | Children in regular foster care aged 0–18 years |
| Randolph [33] | Systematic review | 7 (4) | Q1, Qa, Md | USA = 7 | 2008 - 2015 | Care experienced youth |
| Fry [34] | Systematic review | 31 (3) | Q1 | USA = 18, South America = 4, Canada = 2, Sweden = 2, Israel = 2, UK = 1, Caribbean = 1, Seychelles = 1 | 1995 - 2014 | Young people who have experienced foster care |
| Costa [35] | Systematic review | 25 (25) | Q1 | Spain = 4, Portugal = 7, Israel = 3, Brazil = 2, USA = 1, France = 1, UK = 1, Japan = 1, Iran = 1, Estonia = 1, Korea = 1, Croatia = 1, Romania = 1, | 2000 - 2017 | Adolescents (aged 11–18 years) in residential or institutional care |
| Carr [12] | Systematic review | 49(49) | Md | Austria = 12, Switzerland = 9, Canada = 6, Ireland = 5, USA = 3, Australia = 2, The Netherlands = 1, Germany = 1 | 1996-2018 | Children in long-term care |
| Engler [36] | Systematic review | 25(25) | Q1 | USA = 18, Norway = 3, UK = 1, Germany = 1, Canada = 1, New Zealand = 1 | 2001-2018 | Children who have been in foster care |
| Lou [37] | Systematic review | 15 (15) | Q1, Qa, Md | USA = 2, Singapore = 3, South Africa = 2, Canada = 1, Israel = 1, Croatia = 1, Portugal = 1, Iran = 1, Czech Republic = 1, Greece = 1 | 2002-2017 | Youth (<19 years) in residential care settings |
| DiGiovanni [92] | Literature review | 43 (43) | Q1, Qa, other – reviews | Reported for some studies (includes USA, Australia, North Pacific region) | 1960 - 2019 | Children in foster care |
| DeLuca Bishop [62] | Meta-analysis | 37 (7) | Q1 | USA = 17, Spain = 2, France = 1, Israel = 1 | 1983-2015 | Those with foster care experience |
| Collins [91] | Integrative review | 18 (18) | Q1, Qa | n.r | 2006 - 2015 | Youth (12–30 years of age) in foster care or transitioning out of foster care |

*(Continued)*

| Study | Type of review | Number of studies included (*) | Type of included studies | Location of included studies | Review time frame | Population included |
|---|---|---|---|---|---|---|
| Wilson [38] | Systematic review | 12 (12) | Q1 | USA=3, India=2, Australia=1, Haiti=1, Israel=1, Portugal=1, Spain=1, Turkey=1, Russia=1 | 1989 - 2019 | Children or adolescents (under the age of 20) and drawn from a full-time care setting, with admission due to abuse or neglect at home, parental incapacity, death or inability to provide for the child's needs. |
| Zhang [63] | Meta-analysis | 15 (15) | Q1 | USA=13, Ireland=1 and Switzerland=1. | 2012 - 2020 | Children involved with the child welfare system (including children adopted from the child welfare system). |
| Xu [39] | Systematic review | 8 (8) | Q1 | USA=6, Belgium=1, Norway=1. | 2012 - 2017 | Children or adolescents in kinship or non-kinship foster care |
| Evans [75] | Systematic review & meta-analysis | 5 (5) | Q1 | Canada=2, USA=1, England=1, Australia=1 | 2001 - 2011 | Children and young people that have been placed in care |
| Cameron-Mathiassen [40] | Systematic review | 12(12) | Qa | UK=6, Sweden=2, Norway=1, USA=1, South Africa=1, Australia=1 | 2002-2019 | Children, youths and young adults within the age range 12–25 years, living in residential institutional care of a continuous or ongoing nature. Young people whose residency was not related to physical or learning disabilities |
| Lee [41] | Systematic review | 24(24) | Q1 | USA=24 | 1994-2020 | Children and youth in the USA foster system |
| Thompson [42] | Systematic review | 38 (23) | Q1, Qa, Md | USA=23, n.r | 2006 - 2015 | Older youth in foster care. |
| Hayes [43] | Systematic review | 20 (20) | Qa | UK=9, USA=3, The Netherlands=2, Canada=2, South Africa=1, Denmark=1, Ireland=1 and Sweden=1 | 2003 - 2021 | Children and young people currently living in foster care |
| Kang-Yi [98] | Systematic review | 28 (28) | Q1, Qa, Md | n.r | 1991 - 2013 | Youth with behavioral health disorders aging out of foster care |
| Konijn [64] | Meta-analysis | 42(42) | Q1 | USA/Canada=24, Europe=16, Australia=2 | 1990-2015 | Long-term foster care in Western countries |
| Dubois-Comtois [65] | Meta-analysis | 41 (41) | Q1 | USA=26, England=3, Belgium=2, Australia=1, Canada=1, Chile=1, Croatia=1, Ireland=1, The Netherlands=1, Norway=1, Serbia=1, Spain=1, Turkey=1 | 1988 - 2017 | Children living in foster care at the time of intake, with more than 75% of the children under 21 years of age. |
| Geiger [93] | Literature review | 22 (22) | Q1, Qa, Md | n.r | 2005 - 2015 | Foster care alumni in higher education |
| Kekoni [96] | Integrative literature | 41(41) | Q1, Qa | n.r | 1993-2014 | n.r |
| Li [16] | Meta-analysis | 23 (23) | Q1 | USA=10, UK=2, Serbia=2, Iraqi Kurdistan=1, Croatia=1, Canada=1, The Netherlands=1, Australia=1, Romania=1, South Korea=1, Germany=1, Singapore=1 | 1999 - 2016 | Children in residential care compared to children in family foster care |
| Bergstroöm [76] | Systematic review & meta-analysis | 28 (28) | Q1 | USA=20, The Netherlands=2, UK=1 | 1994 - 2017 | Children up to the age of 17 who are placed in foster family care |
| Biehal [83] | Literature review | 19 (18) | Q1 | USA=11, UK=7, Australia=1 | 1981 - 2008 | Children in foster care |

*(Continued)*

**Table 2.** (Continued)

| Study | Type of review | Number of studies included (*) | Type of included studies | Location of included studies | Review time frame | Population included |
|---|---|---|---|---|---|---|
| Mazzone [97] | Literature review | 31(31) | Q1, Qa | Israel=7, England=5, Spain=2, Croatia=7, Japan=1, The Netherlands=1, Korea=1, Australia=1, n.r=6 | 1997-2017 | Children living in institutional care settings |
| Miranda [84] | Literature review | n.r | n.r | n.r | n.r | Children in foster care and foster care alumni |
| Lund [14] | Scoping review | 25(25) | Q1, Qa, Md | Australia=16, UK=3, International=3, New Zealand=2, Europe=1 | 2010-2020 | Children and young people in out-of-home care in Australia |
| Ãlvarez [13] | Scoping review | 14 (14) | Q1, Qa, Md | USA=14 | 2002-2019 | LGBTQIA+ children and youth in out-of-home care |
| Quiroga [45] | Systematic review | 18 (18) | Q1 | USA=7, Romania=4, Canada=1, Israel=1, Greece=1, Ukraine=1, France=1, Chile=1, Japan=1, Democratic Republic of the Congo=1 | 1993 - 2012 | Children aged 0–17 years living in alternative care for a minimum of 2 months (children's homes and foster care) |
| Boyle [46] | Systematic review | 11 (4) | Q1, Qa | UK=11 | 2002 - 2013 | Children in non-kinship permanent foster care, with formally organized contact mediated by fostering services |
| Eastman [47] | Systematic review | 18(18) | Q1 & Qa | n.r | 2011-2017 | Young parents with CPS involvement |
| Wright [49] | Systematic review | 38 (38) | Q1 | Europe=14, Africa=7, Middle East=4, North America=4, Asia=3, Central America=3, Global=2, South America=1 | 1972 - 2018 | Youth who resided in institutional care between the ages of 0–18 years, and resided full-time within institutional care. |
| Gillum [94] | Literature review | 24 (24) | n.r | n.r | n.r | Individuals who experienced foster care and were college students or college graduates |
| Osei [85] | Rapid review & meta-analysis | 13(13); Review studies (5) | Q1 | USA=13 | 1990 - 2013 | Youths ages 10–18 |
| Stewart [87] | Literature evaluation | 27 (27) | Q1 | USA=17, UK=3, Sweden=3, Australia=1, Canada=1, Germany=1, Norway=1. | 2001 - 2011 | Children and adolescents involved with child welfare services |
| Lutman [15] | Rapid review | 22 (22) | Qa | n.r (but search was limited to UK, Ireland, USA, Canada, New Zealand, Denmark, Norway, Sweden) | 1995-2014 | Children (aged up to 18) living in foster family care (including kinship care) |
| Häggman-Laitila [50] | Systematic review | 13(13) | Qa | USA=9, Israel=1, India=1 | 2010-2017 | Young adults or adults who had been in foster care and had left the care |
| Lloyd [51] | Systematic review | 7 (7) | Q1 | n.r | 2003 - 2012 | Mothers with substance use disorders whose children are is foster care |
| Steels [86] | Critical review | 7(7) | Q1, Qa | Australia=2, UK=1, Ghana=1, Romania=1, Spain=1, Sweden=1 | 2008 - 2015 | Children and workers in residential care homes |
| Rock [52] | Systematic review | 58 (58) | Q1, Qa | UK=22, USA=21, Canada=7, Australia=5, The Netherlands=2, Sweden=1. | 1960 - 2009 | Children in foster-care |
| DeLuca [66] | Meta-analysis | 25(6) | Q1 | USA=7, Canada=1, UK=1 Australia=1, Spain=1, n.r=1 | 1983 - 2012 | Adolescents in foster care |

*(Continued)*

| Study | Type of review | Number of studies included (*) | Type of included studies | Location of included studies | Review time frame | Population included |
|---|---|---|---|---|---|---|
| Brown [90] | Literature review | n.r | n.r | n.r | n.r | Looked after children who are no longer able to live with their parents and reside with extended family members and friends in kin placements compared to looked after children in non-kin care whose placements are provided or purchased directly by the local authority |
| Chodura [77] | Systematic review & meta-analysis | 43(43) | Md | USA = 29, Other = 14 | 1994-2019 | Children in foster care |
| Milde [53] | Systematic review | 6 (3) | Q1 | Sweden = 5, Finland = 1 | 2002-2018 | Former and current child welfare services clients (aged 2–18 years) in Nordic countries |
| Vanderwill [54] | Systematic review | 29(29) | Q1, Qa, Md | USA = 29 | 2003 - 2017 | Children and youth in care. |
| Poon [67] | Meta-analysis | 9(9) | Q1 | USA = 9 | 2010-2019 | Youth currently and/or formerly engaged in the foster care system in the USA. |
| Saarnik [55] | Systematic review | 24 (24) | Q1, Qa, Md | USA = 6, Canada = 4, Sweden = 4, UK = 3, Australia = 3, Spain = 2, Belgium = 1, Germany = 1 | 2000 - 2019 | Foster parents or children |
| Poitras [56] | Systematic review | 18(18) | Q1 | USA = 7, UK = 4, Australia = 2, Canada = 2, Israel = 1, The Netherlands = 1, and Norway = 1 | 1985 - 2018 | Children aged 0–18 years placed in a foster family or in kinship family. |
| Nuñez [57] | Systematic review | 12(12) | Q1 | USA = 12 | 2009-2019 | Foster youth, including foster alumni |
| Hassall [5] | Systematic review & meta-analysis | 31(31) | Q1, Qa | USA, The Netherlands, Sweden, Korea, Israel, Ireland, Australia (Numbers not reported) | 1991-2017 | Children aged 0–18 living in kinship care |
| Seker [78] | Systematic review & meta-analysis | 19 (19) | Q1 | Europe = 10, USA = 8, Oceania = 1 | 1996 - 2019 | Adults with a history of foster care or residential care in childhood or adolescence. |

*Number of studies relevant to our objectives; n.r = not reported; Q1 = quantitative; Qa = qualitative; Md = mixed design

while service access, healthy living behaviours, and general health were primarily associated with individual and system-level factors. Across all indicators, community-level influences were least examined, indicating a consistent gap in the evidence base.

## Physical and legal security

Within the *Physical and Legal Security* domain, several outcome indicators were not examined across all factor levels, and community-level influences were largely absent (Fig 6). Victimisation was mainly discussed in relation to individual child-level factors, while maltreatment and neglect were most often linked to socio-relational influences, with additional attention to individual and system-level factors. General safety and bullying were primarily examined through system-level perspectives. Arrests, referrals, and convictions were discussed in relation to both system-level and socio-relational factors, while incarceration and imprisonment were mainly linked to system-level influences. Offending behaviours were most often associated with system-level and socio-relational factors. Overall, these patterns show a strong emphasis on formal systems and close relationships, with minimal attention to broader community contexts.

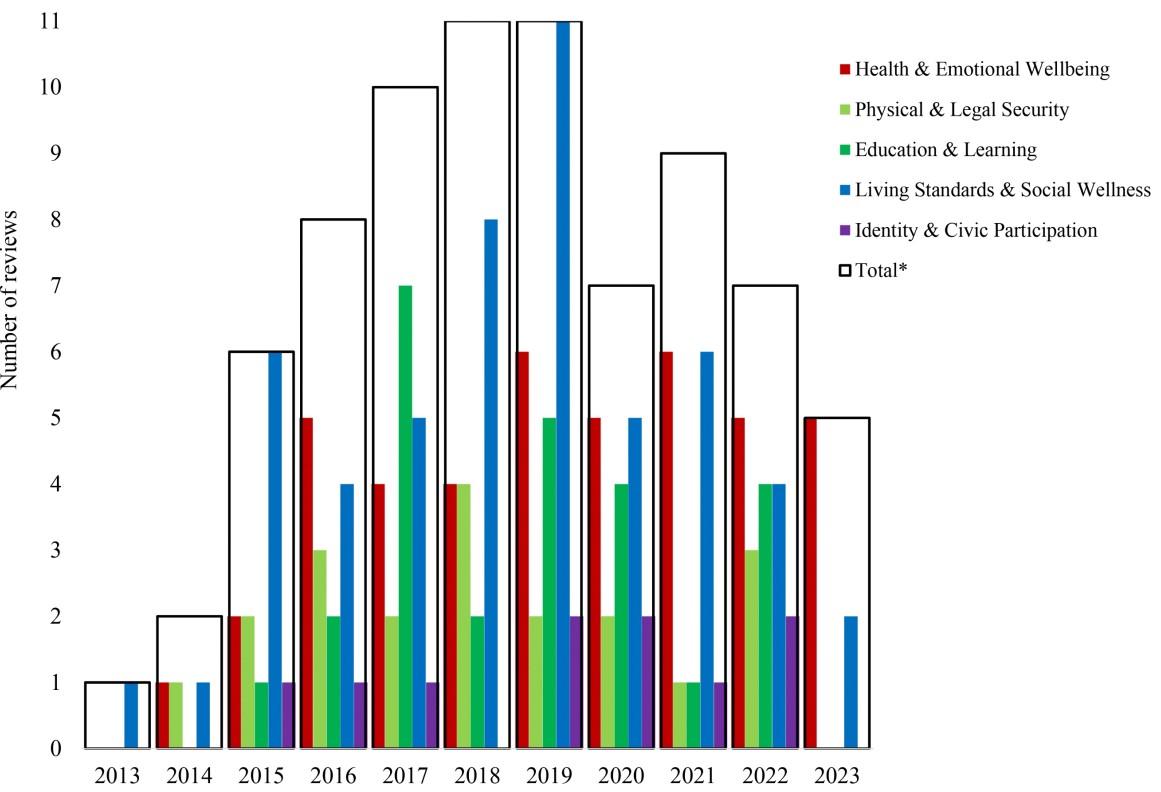

**Fig 2. Trends in the number of review studies on outcomes for children in OHC (2013–2023).** *Since reviews within the same year may cover multiple outcomes, a single review could be counted in several categories, resulting in totals that exceed the number of reviews.

### Education and learning

Within the *Education and Learning* domain, most outcome indicators were examined across multiple factor levels, with the exception of educational readiness and access, which was not linked to other contextual factors (Fig 7). Educational attainment was primarily discussed in relation to individual child and socio-relational influences, while cognitive and academic performance was commonly linked to both individual and system-level factors. School engagement was most often associated with system-level and socio-relational influences. Educational readiness and access was discussed in relation to individual, socio-relational, and system-level factors, but not in relation to other contextual influences. Overall, these patterns indicate that education outcomes in OHC are shaped mainly by individual, relational, and system-level factors, with limited attention to wider contextual influences.

### Living standards and social wellness

Within the *Living Standards and Social Wellness* domain, most outcome indicators were discussed across individual, socio-relational, and system-level factors, while certain indicators showed limited links to broader contexts (Fig 8). Attachment and behaviour functioning were mainly examined in relation to system-level and socio-relational influences, with less attention to community-level factors. Income, deprivation, and poverty were mostly linked to system-level and individual child-level factors. Care-related experiences were primarily associated with socio-relational influences, while employment and labour outcomes were mainly discussed in relation to individual and system-level factors. Adjustment and out-of-care outcomes were examined in relation to both system-level and socio-relational influences. Housing and accommodation were most often linked to system-level factors. Across indicators, community-level and other contextual influences were rarely examined.

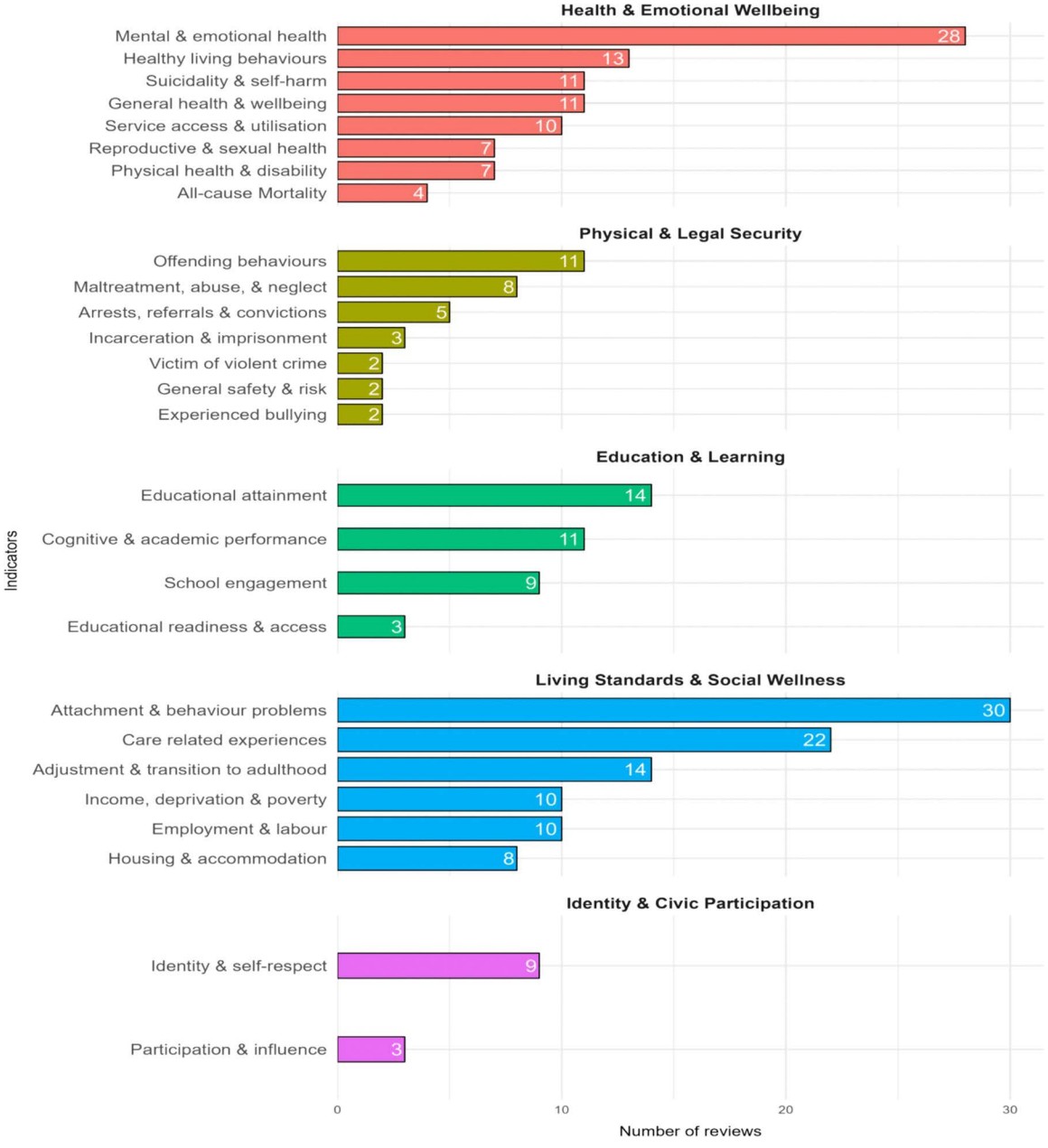

**Fig 3. Number of reviews across outcome domains and indicators.**

### Identity and civic participation

Within the Identity and Civic Participation domain, community-level influences were not reported for any outcome indicators (Fig 9). Identity and self-respect were mainly discussed in relation to socio-relational and other contextual factors, with less attention to system-level and individual child-level influences. Participation and influence were examined only in relation to other contextual and system-level factors. Overall, this domain shows a narrow range of factor-level associations and limited coverage of broader social environments.

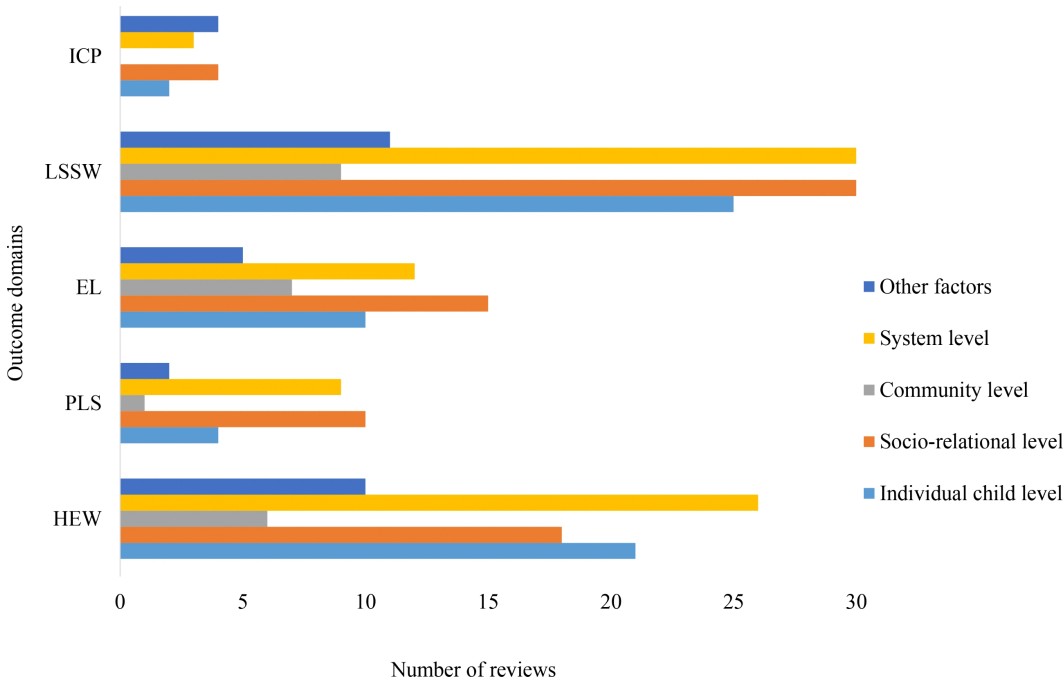

**Fig 4. Frequency of factor reporting across outcome domains.**

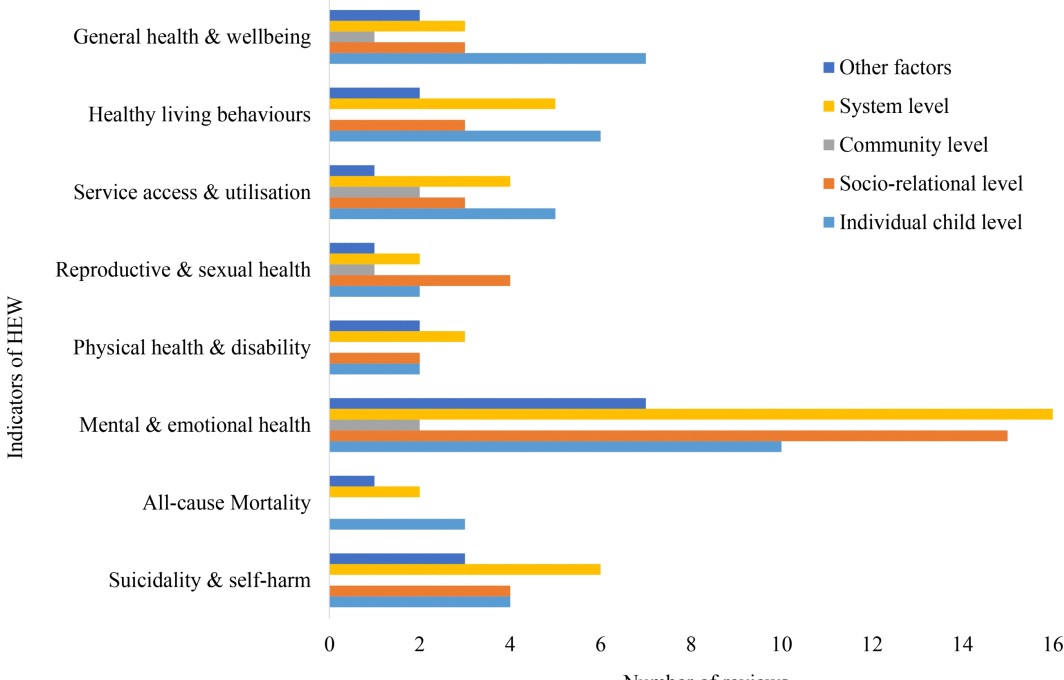

**Fig 5. Distribution of *Health and Emotional Wellbeing* indicators across factor level.**

Fig 6. Distribution of *Physical and Legal Security* indicators across factor levels.

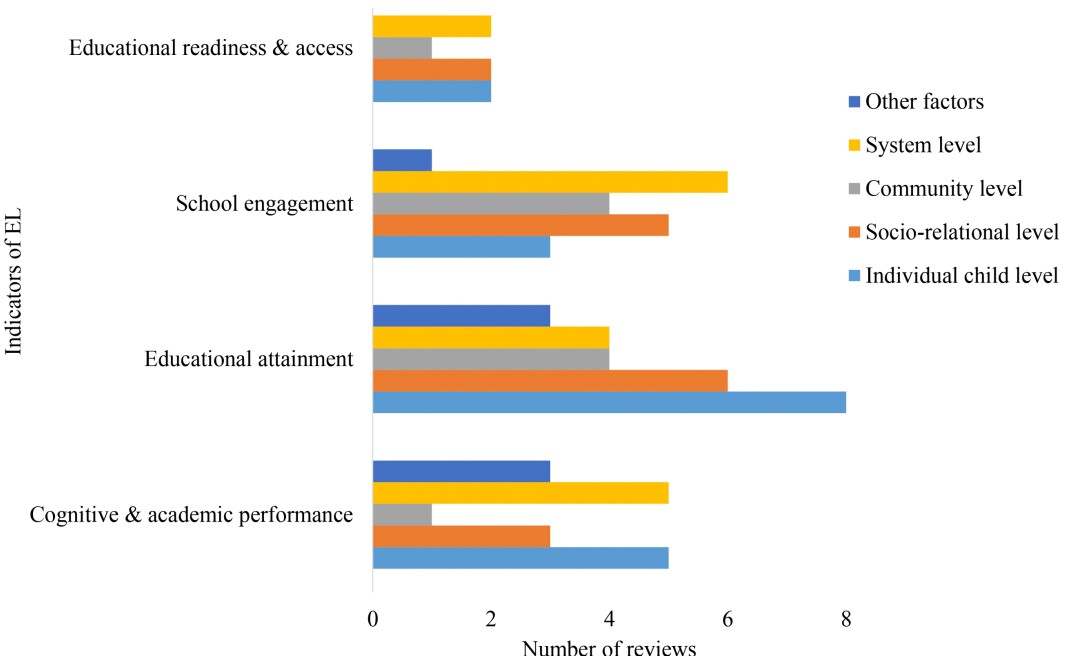

Fig 7. Distribution of *Education and Learning* indicators across factor levels.

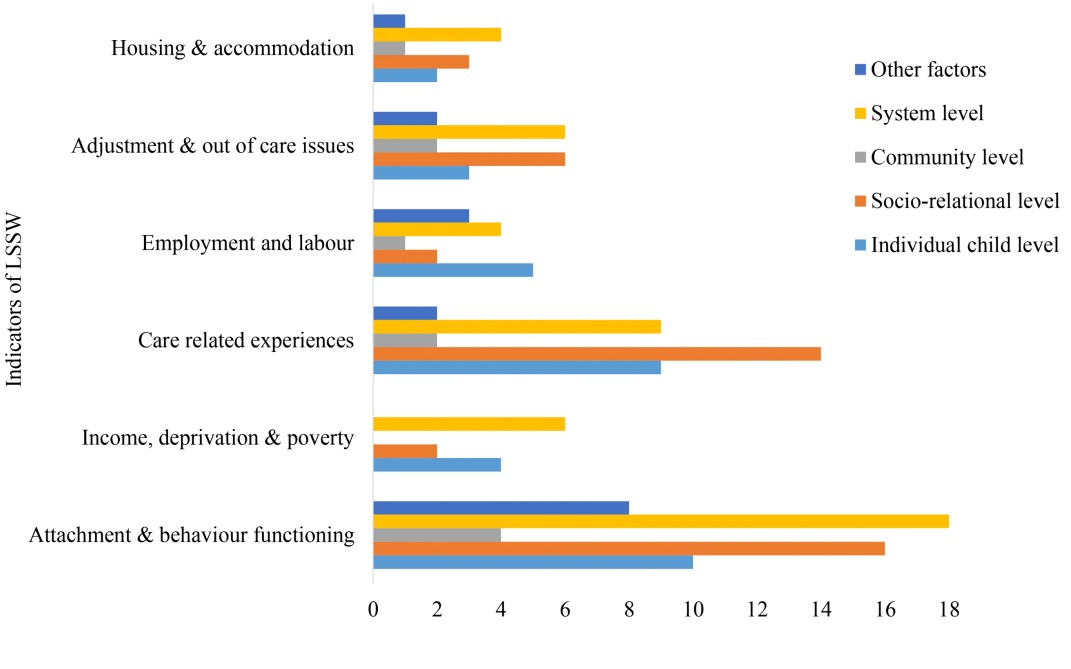

**Fig 8. Distribution of *Living Standards and Social Wellness* indicators across factor levels.**

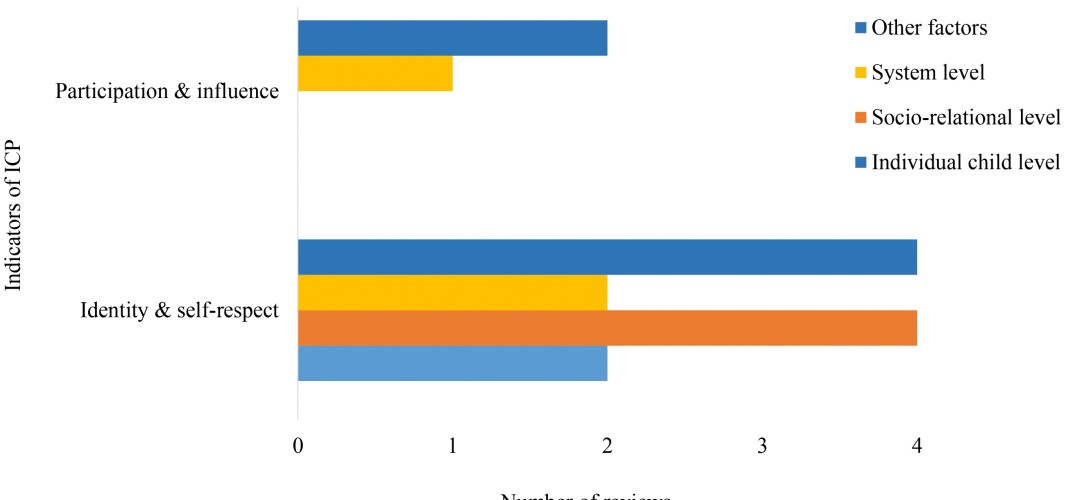

**Fig 9. Distribution of *Identity and Civic Participation* indicators across factor levels.**

## Discussion

This scoping review of 77 reviews identified for the first time critical patterns across domains of *Health and Emotional Wellbeing*, *Physical and Legal Security*, *Education and Learning*, *Living Standards and Social Wellness*, and *Identity and Civic Participation*, offering valuable insights into the factors reported in relation to these outcomes with the potential to

inform future efforts. The novel findings show that research is concentrated in the *Living Standards and Social Wellness* and *Health and Emotional Wellbeing* domains, with fewer reviews on *Identity and Civic Participation* and *Physical and Legal Security*. Most factors were reported at the system, socio relational, and individual child levels, while community level influences were least reported. Viewed within the wider OHC context, these findings show that research attention mirrors long standing priorities in practice, including emotional wellbeing, behaviour, and placement experiences. Areas related to identity, participation, and community conditions have received relatively less focus, although they are important for long term stability and belonging. This suggests that current knowledge may not capture the full range of experiences that shape outcomes for children in OHC.

## Patterns and gaps in research focus

This evidence map demonstrates a concentration of research efforts on specific of care-experienced individuals, particularly *Living Standards and Social Wellness* and *Health and Emotional Wellbeing*, both of which are closely tied to children's immediate and long-term quality of life [13,37]. The consistent volume of research in these areas may reflect their prioritisation in policy and practice. However, *Identity and Civic Participation* emerged as the least reported domain, with far fewer reviews addressing outcomes about identity development, self-respect, influence and community participation. This disparity might suggest a critical gap in primary research, particularly given the importance of identity development and civic participation in fostering a sense of belonging and resilience among care-experienced children and young people [15]. Research on indicators often reflected uneven attention across domains. For example, within *Health and Emotional Wellbeing*, substantial emphasis was found on mental and emotional health, while indicators such as all-cause mortality had received little attention. A similar pattern was observed in *Physical and Legal Security*, where offending behaviours were well documented, whereas indicators such as bullying and being a victim of violent crime were relatively underrepresented. These patterns suggest a need for more balanced research efforts to ensure that less reported but equally critical aspects of children's experiences are adequately addressed.

The review highlighted the prominence of system level factors (e.g., placement types and care quality) across all outcome domains. These factors were most frequently reported in relation to indicators such as attachment and behaviour functioning, as well as mental and emotional health. This finding aligns with a previous correlates review [98], highlighting the significant role of system and carer-related factors, particularly their impact on placement stability and behaviour of children in out-of-home care (OHC). However, while system-level factors were well-represented, community factors were minimally reported across all outcome domains. This gap may point to an unmet need for research exploring the influence of community-level factors such as neighbourhood characteristics in influencing outcomes of children in OHC.

The findings of this review of reviews also highlight the consistent reporting of socio-relational factors across several outcome indicators. Supportive social relationships with family and peers often act as protective factors, showing the value of maintaining family ties and supportive relationships for care-experienced children [43,55,70,96]. Nonetheless, the underrepresentation of socio-relational factors in domains such as *Identity and Civic Participation* and *Legal and Physical Security* points to potential areas for further exploration.

## Strengths and limitations

This review represents a comprehensive analysis of review-level evidence, providing an overview of outcomes and associated factors for children in OHC. By extracting data from full texts rather than just abstracts, this study minimised the risk of missing relevant outcomes and factors. The involvement of care-experienced people in the study strengthened the review by incorporating diverse perspectives in refining the search strategy and contextualising findings [99]. However, limitations include the variability in methodological approaches and scope across the included reviews, which may affect the generalisability of some findings. The reliance on descriptive analysis, while useful for mapping the evidence base, limits the depth of the review's conclusions. Nonetheless, the data and specific factor-outcome relationships are provided

as a resource for stakeholders seeking evidence on factors influencing outcomes for children in OHC (see S2 File). It is worth noting that some included reviews [44,61,66,75,78] did not report moderating variables influencing outcomes but focused solely on outcomes for children following placement in OHC. These reviews typically compared children in OHC with those in the general population, offering limited insight into factors shaping various outcomes within the care-experienced group.

### Future directions

Future research should aim to address the identified gaps by focusing on underexplored domains, such as *Identity and Civic Participation* and indicators including bullying, mortality, and educational readiness. Policymakers, practitioners, and researchers should collaborate to ensure that commissioning and evaluation frameworks place greater emphasis on community-level influences which were minimally reported across reviews despite their potential relevance to outcomes for children in OHC. Methodologies such as participatory research, involving care-experienced children and young people, can provide deeper insights into their lived experiences and priorities.

### Conclusion

This review of reviews highlights significant patterns and gaps in the research on outcomes for children in OHC. We have not presented a synthesis of results on the direction or strength of associations and therefore are not making claims about causality. However, the paper does present a comprehensive overview of risk and protective factors and outcomes that have been measured to date. These are aspects that professionals should be aware of in their practice and consider in their assessments and ongoing progress monitoring of children and their families. Where evidence gaps exist, services should consider collecting high-quality data routinely, for example, by asking children and young people about their civic participation and developing identities. Such routine data could then be analysed through collaboration with researchers.

### Supporting information

**S1 Table. Development of outcome domains and indicators.**
(DOCX)

**S2 File. Factors reviewed on various outcome indicators.**
(XLSX)

**S3 Table. PRISMA checklist extension for scoping reviews.**
(DOCX)

### Author contributions

**Conceptualization:** Richmond Opoku, Natasha Judd, Katie Cresswell, Michael Parker, Jonathan Scourfield, Karen Hughes, Jane Noyes, Dan Bristow, Evangelos Kontopantelis, Sinead Brophy, Natasha Kennedy.

**Data curation:** Richmond Opoku, Natasha Judd, Katie Cresswell, Michael Parker, Michaela James, Jonathan Scourfield, Evangelos Kontopantelis, Sinead Brophy, Natasha Kennedy.

**Formal analysis:** Richmond Opoku, Natasha Judd, Katie Cresswell, Michael Parker, Karen Hughes, Natasha Kennedy.

**Methodology:** Richmond Opoku, Natasha Judd, Katie Cresswell, Michael Parker, Karen Hughes, Jane Noyes, Dan Bristow, Evangelos Kontopantelis, Sinead Brophy, Natasha Kennedy.

**Supervision:** Michaela James, Jonathan Scourfield, Karen Hughes, Jane Noyes, Dan Bristow, Evangelos Kontopantelis, Sinead Brophy, Natasha Kennedy.

**Validation:** Natasha Judd, Michaela James, Karen Hughes, Sinead Brophy.

**Visualization:** Richmond Opoku.

**Writing – original draft:** Richmond Opoku.

**Writing – review & editing:** Richmond Opoku, Natasha Judd, Katie Cresswell, Michael Parker, Michaela James, Jonathan Scourfield, Karen Hughes, Jane Noyes, Dan Bristow, Evangelos Kontopantelis, Sinead Brophy, Natasha Kennedy.

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
