## [Editor Report · Decision Letter 0]

19 Jul 2025

PONE-D-25-25545

Mapping the Evidence on Outcomes of Childhood Out-of-Home Care: A Scoping Review of Reviews

PLOS ONE

Dear Dr. Opoku,

Thank you for submitting your manuscript to PLOS ONE. After careful consideration, we have decided that your manuscript does not meet our criteria for publication and must therefore be rejected.

Specifically:

The scoping review of reviews is outside the scope for PLOS ONE. The link you have provided to your protocol is invalid and registration status is not able to be confirmed.

I am sorry that we cannot be more positive on this occasion, but hope that you appreciate the reasons for this decision.

Kind regards,

Janna Metzler

Academic Editor

PLOS ONE

- - - - -

---

## [Author Response · Author response to Decision Letter 1]

31 Jul 2025

Response to the Editor’s comments

Editor’s Comments

Thank you for submitting your manuscript to PLOS ONE. After careful consideration, we have decided that your manuscript does not meet our criteria for publication and must therefore be rejected.

Specifically:

The scoping review of reviews is outside the scope for PLOS ONE. The link you have provided to your protocol is invalid and registration status is not able to be confirmed.

Authors’ Response

We are grateful for your time and the opportunity to submit to PLOS ONE.

Firstly, it seems that our submission was considered out of scope not because of the topic—childhood out‑of‑home care—but rather because of our chosen methodology, namely a scoping review of reviews. However, this interpretation appears difficult to reconcile with PLOS ONE’s stated aims and the following recent examples of scoping reviews‑of‑reviews published in the journal:

• Craske et al. (2024), Pharmacist-led medication reviews: A scoping review of systematic reviews, published 06 Sep 2024

https://doi.org/10.1371/journal.pone.0309729

• Breslin et al. (2024), Whole systems approaches to diet and healthy weight: A scoping review of reviews, published 13 Mar 2024

https://doi.org/10.1371/journal.pone.0292945

• Hypotheses and evidence related to intense sweeteners and effects on appetite and body weight changes: A scoping review of reviews, published 18 July, 2018

https://doi.org/10.1371/journal.pone.0199558

If there has not been a recent change in editorial policy that excludes scoping review of reviews then we think our submission is within the scope of PLOS ONE.

Secondly, we reviewed our submission in the PLOS ONE Editorial Manager system, and the protocol link embedded in the submission proof is valid and accessible on our end. However, if the link did not work for the editorial team, we would have appreciated the opportunity to rectify this before a final decision was made. For your convenience, we have reattached the protocol link here: https://doi.org/10.17605/OSF.IO/G7D5J

We fully appreciate the volume of submissions the journal handles and understand editorial discretion is exercised in line with scope and scientific rigor. However, given the above, we respectfully request reconsideration if appropriate, and would be grateful for any direction you may offer on next steps.

Thank you for your attention, and we look forward to your kind response.

Warm regards,

Richmond Opoku

[On behalf of the authors]

---

## [Decision Letter · Decision Letter 1]

30 Oct 2025

Dear Dr. Opoku,

Thank you for submitting your manuscript to PLOS ONE. After careful consideration, we feel that it has merit but does not fully meet PLOS ONE’s publication criteria as it currently stands. Therefore, we invite you to submit a revised version of the manuscript that addresses the points raised during the review process.



We look forward to receiving your revised manuscript.

Kind regards,

Mary Diane Clark, PhD

Academic Editor

PLOS ONE

Journal Requirements:

2. Thank you for stating the following financial disclosure: [This research was funded by the National Institute for Health Research (NIHR156826 - CARELINK Wales - Comprehensive Analysis of Risk factors and outcomes for vulnerable children through LINKed Welsh data), UK, and the Economic and Social Research Council – Administrative Data Research (ESRC-ADR), UK (PhD studentship).].

3. We note that there is identifying data in the Supporting Information file <S2 Table. Study characteristics.docx and S3 File. Factors reviewed on various outcome indicators.xlsx>. Due to the inclusion of these potentially identifying data, we have removed this file from your file inventory. Prior to sharing human research participant data, authors should consult with an ethics committee to ensure data are shared in accordance with participant consent and all applicable local laws.

-Location data

Please remove or anonymize all personal information (Name), ensure that the data shared are in accordance with participant consent, and re-upload a fully anonymized data set. Please note that spreadsheet columns with personal information must be removed and not hidden as all hidden columns will appear in the published file.

4. Please amend your list of authors on the manuscript to ensure that each author is linked to an affiliation.

We note that you have included affiliation numbers 1,2,3 and 4 however only affiliations 1,2 and 3 have authors linked to them. Please amend affiliation 4 to link an author to it or remove if added in error.

4, Please include captions for your Supporting Information files at the end of your manuscript, and update any in-text citations to match accordingly. Please see our Supporting Information guidelines for more information: http://journals.plos.org/plosone/s/supporting-information.

Additional Editor Comments (if provided):

Thank you for submitting your manuscript to PLOS ONE. I was only able to obtain one review but in terms of time, I will act as the second. Most of my concerns are expressed by the one review so please respond to all of those. You have some issue with your time line---in one eligibility you state that the time line is 2003 – 2024. Then in the results you say 1972-2022. Need to clean up the procedure.

You have many graphs but then you restate those same items with the number of papers found in the text. The rule is one place or the other---so i suggest stay with the tables and graphs and reduce the text to information not available in the graphs.

Finally, the reviewer notes where you need to include all of the procedures in the paper not in the supplemental materials--I agree. The content feels like a sketch rather than a painting --- so it lacks depth. Please expand how you present the data to aid in providing more details.

Reviewers' comments:

Reviewer's Responses to Questions

**Comments to the Author**

Reviewer #1: (No Response)

2. Is the manuscript technically sound, and do the data support the conclusions?

Reviewer #1: Yes

3. Has the statistical analysis been performed appropriately and rigorously?

Reviewer #1: N/A

4. Have the authors made all data underlying the findings in their manuscript fully available?

Reviewer #1: Yes

5. Is the manuscript presented in an intelligible fashion and written in standard English?

Reviewer #1: Yes

Reviewer #1: Greetings Authors,

Thank you for taking the time to revise this manuscript. It is my first time reviewing them, so bear with my comments as I help you improve the manuscript before it is ready for publication. This manuscript aims to provide a bird's-eye view of the reviews that were done related to children who experienced OHC. However, please ensure you define OHC. The authors cannot assume the reader knows what it means, especially in the PLOS ONE journal, which covers a wide range of research.

Now to the substantial part of the manuscript, acronyming the domains carries the risk of overloading the reader’s cognitive abilities, as they have to refresh what each domain stands for. I would recommend the authors to write the factors out instead of using the acronym, such as health and emotional well-being, instead of HEW. Regarding the domains, how did your team determine that the review falls under that factor? It will also be helpful to readers to know the process of coding domains to reviews, because Figure 1 initially showed more factors than the total number of reviews. The footnote does not seem sufficient to describe how the authors determined which factors to assign to the review. That will need to be clear in the method section.

Table 1 should list all indicators rather than selecting example ones; this will help readers orient themselves before reading the results, rather than realizing there are many indicators when they reach Figure 3.

Why do Figures 4 & 5 show the inverse list of factors? Keep in mind to reduce the cognitive overload for the readers to improve the reader’s appreciation of your work. The axis is also inverse too. Pick one format and stick with it throughout all charts to minimize the cognitive overload.

Checking out the protocol in OSF, why is the information in OSF not included in the manuscript, such as the background and research questions? The authors should not assume that the readers will take the time to review the protocol in OSF. Again, to minimize the cognitive overload, the authors should have this information in the manuscript.

There should be a paragraph at the end of the results synthesizing the findings. What does the result mean to the broader audience? How do they answer your research questions? The discussion recapped some of the results but needs to make a more concrete effort to connect them to the broader context of OHC and to offer recommendations for potential future research.

Regarding on the writing and mechanism, it is mostly in a good shape except for few typos and paragraph formatting. The authors will need to proof them before the next submission since PLOS One does not have a copyeditor who could clean up the manuscript before publishing.

**Do you want your identity to be public for this peer review?** For information about this choice, including consent withdrawal, please see our Privacy Policy

Reviewer #1: **Yes:** Scott Cohen

---

## [Author Response · Author response to Decision Letter 2]

21 Dec 2025

Dear reviewers,

We sincerely thank you for your careful reading of our manuscript and for your constructive comments. We have revised the manuscript in response to all suggestions raised. For ease of reference, we reproduce each reviewer comment below, followed immediately by our corresponding response in yellow highlighted text.

Reviewer 1 (Editor)

The information has potential so please reduce the redundancy between graphs and texts.

We have revised the Results section to reduce redundancy between the figures and the accompanying text. Specifically, we removed repeated numerical descriptions that are already clearly displayed in the figures and retained only concise interpretive statements in the text. The revised Results now focus on summarising overall patterns and contrasts across domains, while the detailed distributions are presented in the figures.

I find the manuscript to be a bit sketchy and it would be better to have more details as noted in the reviewers comments.

We have addressed this concern by adding further detail and clarification across the manuscript in line with the reviewers’ suggestions. Specifically, we expanded the methodological explanations, strengthened the synthesis and interpretation in the Discussion, and provided additional contextual detail where necessary. In addition, we have included a paragraph on Public involvement in the methods section and Study characteristics in the results section.

There is one statement about family being the most important place---can you clarify this statement? If a child is placed out of the home it most often suggests that the home for that child is not the best place.

We did not intend to suggest that the home environment is always the best place for every child, particularly in cases where children are placed out of home due to safety or welfare concerns. What we sought to convey is that supportive relationships with family members and peers can play a protective role for care-experienced children. We have clarified this wording to better reflect this distinction, as shown in lines 424–426 of the revised manuscript.

Reviewer 2

Thank you for taking the time to revise this manuscript. It is my first time reviewing them, so bear with my comments as I help you improve the manuscript before it is ready for publication. This manuscript aims to provide a bird's-eye view of the reviews that were done related to children who experienced OHC. However, please ensure you define OHC. The authors cannot assume the reader knows what it means, especially in the PLOS ONE journal, which covers a wide range of research.

We agree that a clear definition of out-of-home care (OHC) is important for readers across disciplines. We have now included a concise definition in the Background at lines 91–94 of the revised manuscript.

Now to the substantial part of the manuscript, acronyming the domains carries the risk of overloading the reader’s cognitive abilities, as they have to refresh what each domain stands for. I would recommend the authors to write the factors out instead of using the acronym, such as health and emotional well-being, instead of HEW.

We have revised the manuscript to write out the domain names in full throughout the text to improve readability and reduce cognitive load for the reader. The only exception is Figure 4, where acronyms are retained due to formatting constraints. All acronyms are clearly defined to support interpretation.

Regarding the domains, how did your team determine that the review falls under that factor? It will also be helpful to readers to know the process of coding domains to reviews, because Figure 1 initially showed more factors than the total number of reviews. The footnote does not seem sufficient to describe how the authors determined which factors to assign to the review. That will need to be clear in the method section.

We have added further detail to the Methods section to clarify how outcome domains and factor levels were assigned to each review, including how reviews could be mapped to multiple domains and factors when applicable. These additional explanations address why the frequency of domains could exceed the total number of reviews and outline the process used for coding. The clarifications are provided in lines 176–180 and 183–196 of the revised manuscript.

Table 1 should list all indicators rather than selecting example ones; this will help readers orient themselves before reading the results, rather than realizing there are many indicators when they reach Figure 3.

We have revised Table 1 to include all outcome indicators rather than a selection of examples. This change is intended to help readers orient themselves to the full scope of indicators before engaging with the Results section.

Why do Figures 4 & 5 show the inverse list of factors? Keep in mind to reduce the cognitive overload for the readers to improve the reader’s appreciation of your work. The axis is also inverse too. Pick one format and stick with it throughout all charts to minimize the cognitive overload.

To improve consistency and reduce cognitive load for readers, we have reformatted Figure 4 to match the factor ordering and axis orientation used in Figure 5 and all subsequent figures.

Checking out the protocol in OSF, why is the information in OSF not included in the manuscript, such as the background and research questions? The authors should not assume that the readers will take the time to review the protocol in OSF. Again, to minimize the cognitive overload, the authors should have this information in the manuscript.

We agree that key information from the OSF protocol should be clearly presented within the manuscript itself. We have now explicitly included the research question in the Background (lines 107–110). We also note that the registered project generated two related papers, one of which has already been published. The current manuscript addresses one specific component of the broader protocol. In line with PLOS ONE guidelines, the related published paper has been cited in this manuscript (see line 129) and was also uploaded as an additional file for peer review purposes to ensure full transparency.

There should be a paragraph at the end of the results synthesizing the findings. What does the result mean to the broader audience? How do they answer your research questions? The discussion recapped some of the results but needs to make a more concrete effort to connect them to the broader context of OHC and to offer recommendations for potential future research.

To avoid redundancy between the Results and Discussion sections, we integrated the requested synthesis at the beginning of the Discussion (see lines 383–392). We have also strengthened the Discussion to offer clearer implications and directions for future research. These revisions summarise the key findings, explains how they address the research question, and positions the results within the broader OHC context.

Regarding on the writing and mechanism, it is mostly in a good shape except for few typos and paragraph formatting. The authors will need to proof them before the next submission since PLOS One does not have a copyeditor who could clean up the manuscript before publishing.

We have carefully proofread the entire manuscript to correct typographical errors and improve paragraph formatting in preparation for the revised submission.

---

## [Decision Letter · Decision Letter 2]

26 Jan 2026

Mapping the Evidence on Outcomes of Childhood Out-of-Home Care: A Scoping Review of Reviews

PONE-D-25-25545R2

Dear Dr. Opoku,

We’re pleased to inform you that your manuscript has been judged scientifically suitable for publication and will be formally accepted for publication once it meets all outstanding technical requirements.

Kind regards,

Mary Diane Clark, PhD

Academic Editor

PLOS One

Additional Editor Comments (optional):

Thank you for the changes. The solve the issues noted earlier. I do have one comment/issue. Some of your tables are not publication perfect. I am asking the journal to check and make that decision. I think the tables are great but don't seem to be the correct format for printing.

Reviewers' comments:

Reviewer's Responses to Questions

**Comments to the Author**

Reviewer #1: All comments have been addressed

2. Is the manuscript technically sound, and do the data support the conclusions?

Reviewer #1: Yes

3. Has the statistical analysis been performed appropriately and rigorously?

Reviewer #1: N/A

4. Have the authors made all data underlying the findings in their manuscript fully available?

Reviewer #1: Yes

5. Is the manuscript presented in an intelligible fashion and written in standard English?

Reviewer #1: Yes

Reviewer #1: Greetings authors,

Thank you for revising the manuscript. The readability has dramatically improved as I read it. I appreciated the Tables, as they helped me understand your findings and discussion. It made your review more straightforward and to the point for future research to identify which research areas warrant more urgent attention, such as identity and civic participation. I have not identified any errors and believe it is ready for publication.

**Do you want your identity to be public for this peer review?** For information about this choice, including consent withdrawal, please see our Privacy Policy

Reviewer #1: **Yes:** Scott Cohen

---

## [Editor Report · Acceptance letter]

PONE-D-25-25545R2

PLOS One

Dear Dr. Opoku,

I'm pleased to inform you that your manuscript has been deemed suitable for publication in PLOS One. Congratulations! Your manuscript is now being handed over to our production team.

Kind regards,

on behalf of

Dr. Mary Diane Clark

Academic Editor

PLOS One